# Kaempferol as a Dietary Anti-Inflammatory Agent: Current Therapeutic Standing

**DOI:** 10.3390/molecules25184073

**Published:** 2020-09-07

**Authors:** Waqas Alam, Haroon Khan, Muhammad Ajmal Shah, Omar Cauli, Luciano Saso

**Affiliations:** 1Department of Pharmacy, Abdul Wali Khan University Mardan, Mardan 23200, Pakistan; waqasalamyousafzai@gmail.com; 2Department of Pharmacognosy, Faculty of Pharmaceutical Sciences, Government College University, Faisalabad 38000, Pakistan; ajmalshah@gcuf.edu.pk; 3Department of Nursing, University of Valencia, 46010 Valencia, Spain; omar.cauli@uv.es; 4Department of Physiology and Pharmacology “Vittorio Erspamer”, Sapienza University, P.le Aldo Moro 5, 00185 Rome, Italy

**Keywords:** flavonoids, kaempferol, inflammation, therapeutic utility

## Abstract

Inflammation is a physiological response to different pathological, cellular or vascular damages due to physical, chemical or mechanical trauma. It is characterized by pain, redness, heat and swelling. Current natural drugs are carefully chosen as a novel therapeutic strategy for the management of inflammatory diseases. Different phytochemical constituents are present in natural products. These phytochemicals have high efficacy both in vivo and in vitro. Among them, flavonoids occur in many foods, vegetables and herbal medicines and are considered as the most active constituent, having the ability to attenuate inflammation. Kaempferol is a polyphenol that is richly found in fruits, vegetables and herbal medicines. It is also found in plant-derived beverages. Kaempferol is used in the management of various ailments but there is no available review article that can summarize all the natural sources and biological activities specifically focusing on the anti-inflammatory effect of kaempferol. Therefore, this article is aimed at providing a brief updated review of the literature regarding the anti-inflammatory effect of kaempferol and its possible molecular mechanisms of action. Furthermore, the review provides the available updated literature regarding the natural sources, chemistry, biosynthesis, oral absorption, metabolism, bioavailability and therapeutic effect of kaempferol.

## 1. Introduction

Flavonoids are the secondary metabolites of plants which are characterized by polyphenolic structures. They are a class of natural products and are richly distributed in vegetables, fruits and some kinds of beverages. Flavonoids have biochemical, anti-inflammatory, antioxidant and anti-mutagenic effects that are responsible for the management of various ailments like atherosclerosis, cancer and Alzheimer’s disease [1,2]. Flavonoids possess a wide range of health-promoting properties and are indispensable components in various pharmaceutical, cosmetics, nutraceutical and medicinal applications. This is due to the ability to modulate the functions of cellular key enzymes as well as the antioxidant effect, anti-carcinogenic and anti-inflammatory effect of flavonoids. Flavonoids inhibit different enzymes like lipo-oxygenase, cyclo-oxygenase, phosphoinositide 3-kinase and xanthine oxidase [1,3].

Flavonoids are chemicallylow-molecular weight products of natural plants and are extracted and isolated from the different parts. They are abundantly found in all parts of the plants, foods and some beverages of plant origin. Flavonoids are further divided into subgroups on the basis of the structure of their carbon rings and these divisions include flavanols, flavones, chalcones, flavnones, flavanonols and isoflavones. Flavonoids are major contributors to the color and fragrance of fruits and flowers which helps in pollination [4,5]. Flavonoids have been reported to have potent anti-inflammatory effects. It has been shown that flavonoids are involved in the inhibition of various enzymes which provoke the inflammation process [6,7,8]. Chemical mediators like prostaglandins and nitric oxide production are involved in the process of inflammation [9].

A wide range of flavonoids have been isolated from plants such as luteolin, nobiletin, tangeretin, apigenin, sinensetin, quercetin and kaempferol. Flavonols are the sub-group and building blocks of proanthocyanines and of flavonoids which possess a ketone group in their structures. Flavones have a hydroxyl group in their structure at carbon 3 of the central oxygen containing 6-member ring. The flavones include quercetin, fisetin, myricetin and kaempferol. They are abundantly found in plants and vegetables like lettuce, grapes, tomatoes, onion, berries, kale and also in wine and tea [10,11]. The aim of this review is to comprehensively summarize the sources, chemistry and pharmacological effects of kaempferol, majorly focusing on the anti-inflammatory effect.

## 2. Chemistry of Kaempferol

Kaempferol (3,5,7-trihydroxy-2-(4-hydroxyphenyl)-4*H*-chromen-4-one) is a polyphenol which is richly found in fruits and vegetables (Figure 1). It is also found in plant-derived beverages. Kaempferol is used in the management of various ailments but there is no available review article that can summarize all the natural sources and biological activities specifically focusing on the anti-inflammatory effects of kaempferol.

## 3. Natural Sources of Kaempferol

Kaempferol is widely distributed in foods, beverages and the plant kingdom. Kaempferol and its derivatives are synthesized in plants by different types of enzymes. Different dietary sources of kaempferol are summarized in Table 1. It has also been found that kaempferol is richly found in Magnoliophyta, Pinophyta, Pteridophyta, Aspidiaceae, Polypodiaceae and Aspleniaceae (Appendix A).

## 4. Isolation of Kaempferol from Natural Sources

Phytochemical screening of different medicinal plants has confirmed the presence of flavonoids like kaempferol and its derivative compounds.

Further screening and fractionation were done to isolate kaempferol and its derivatives. Orhan and his coworkers performed successive column chromatography techniques for bioactive-guided fractionation of *Calluna Vulgaris* and isolated kaempferol galactoside [17].

Yang and his coworkers fractionated the dried roots of *Neocheiropteris palmatopedata*. They purified and separated kaempferol and its derivatives from the methanolic crude extract of *N. palmatopedata* by repeated column chromatography, using a Sephadex LH-20 column [18]. Liang and his co-researchers used high-speed countercurrent chromatography to isolate and purify kaempferol and its compounds from the leaf extract of *Siraitiagrosvenori.* They used 90 mg of the plant extract in a two-phase solvents system. Yang and his coauthors isolated kaempferol-3,7-O-a-L-dirhamnopyranoside from a plant extract and its structure was identified using ^1^H-NMR, MS and ^13^C-NMR [19].

## 5. Biosynthesis of Kaempferol

Kaempferol is structurally composed of diphenylpropane, which is prepared via condensation by the help of different enzymes (Figure 2). These enzymes are very common and abundantly found in plants. In the condensation process, three molecules of malonyl-CoA and one molecule of 4-coumaroyl-CoA are used. Chalcone synthase is responsible for the condensation process which results in the formation of the flavonoid naringenin chalcone. The enzyme chalcone isomerase further converts the naringenin chalcone into naringenin flavanone by closing the C3 ring. A hydroxyl group is added to the C3 ring of naringenin using the catalytic activity of theflavanone-3-dioxygenase enzyme and as a result, dihydrokaempferol is formed. In the last step, a double bond is introduced at C2–C3 of dihydrokaempferol by flavonol synthase [20,21,22].

## 6. Bioavailability, Oral Absorptionand Metabolism of Kaempferol

To establish the pharmacokinetic parameters of a newly isolated compound, both in vivo and in vitro studies are performed. Similarly, for kaempferol, numerous in vivo and in vitro studies were done to determine the pharmacokinetic parameters. As kaempferol is a dietary flavonoid and for its therapeutic effects, it is necessary for it to be absorbed, distributed, metabolized and eliminated properly. Kaempferol is usually ingested orally as high-polarity and low-polarity glycosides. The high-polarity glycosides show resistance while low-polarity glycosides are easily absorbed. Kaempferol is lipophilic in nature and like other flavonoids, kaempferol is also absorbed from the small intestine.

It has been reported that kaempferol is absorbed by passive diffusion, facilitated diffusion as well as by active transport due its lipophilicity. Kaempferol is metabolized by a glucuronide conjugation as well as by a sulfate conjugation in the liver [23,24,25]. Kaempferol is also metabolized in the small intestine by intestinal enzymes. The normal floras of the colon metabolizes the kaempferol glycoside into aglycones and further convert it into 4-methylphenol, 4-hydroxyphenylacetic acid and phloroglucinol which are absorbed into the systemic circulation and then distributed to different tissues and finally excreted via feces or urine [26,27]. It has been reported that about 1.9% to 2.5% of the ingested kaempferol was excreted in urine [28,29].Cao and his co-researchers investigated the bioavailability and pharmacokinetics of kaempferol in humans and found that the kaempferol plasma concentration was 57.86 nM upon intake of 14.97 mg/day. It was also concluded that a plasma concentration of 15 ng/ml kaempferol was detected from 27 mg kaempferol obtained from tea [29].

It has been concluded from several studies that the oral bioavailability of kaempferol is in the nano- or microgram per ml range but still it has wide therapeutic effects and is used for the management of different ailments. Thus, it might be assumed that the kaempferol is therapeutically active in minute doses or there might be some active metabolites produced upon the metabolism of kaempferol which will act as therapeutic agents.

## 7. Anti-Inflammatory Effect of Kaempferol

Inflammation is a physiological response to different pathological, cellular or vascular damages due to physical, chemical or mechanical trauma. Inflammation is a protective mechanism that protects the living organism against tissue injury. Redness, pain, loss of function and heat are the signs associated with inflammation [30,31,32]. Inflammation may be acute or chronic. Inflammation is caused due to the activation of the immune system. Inflammation activates numerous immune cells such as neutrophils, B-cells, T-cells and NK cells. The immune response and inflammatory process is initiated by different enzymes like cyclo-oxygenase, lipo-oxygenase, protein kinase, phospholipase A, Phosphodiesterase and tyrosine kinase [33,34]. These enzymes activate the endothelial cells and play a key role in inflammatory responses. Phosphorylation by protein kinases and nitric oxide which is formed by nitric oxide synthase plays a crucial role in the inflammatory process. Furthermore, cyclo-oxygenase activates the synthesis of prostaglandins and prostacyclin via the arachidonic acid pathway and activates the immune response in the inflammatory process [35,36,37]. It has been reported by different researchers that kaempferol has significant anti-inflammatory effects. Kaempferol exerts its anti-inflammatory effects via different mechanisms (Table 2) which are summarized below.

### 7.1. In Vitro Studies

The cyclooxygenase pathway produces prostaglandins which are mediators of inflammation. Nitric oxide synthase accelerates the synthesis of nitric oxide at the site of inflammation which further augments the synthesis of prostaglandins [42,43].Lipo-oxygenase is involved in the synthesis of leukotrienes that cause different diseases like inflammatory bowel disease, asthma and rheumatoid arthritis [44].

The in vitro studies have shown that kaempferol has a significant inhibitory effect on COX1 and COX2.The cyclooxygenase pathway is activated when a physical, chemical or mechanical trauma occurs at any site of the body. The phospholipids’ membranes are ruptured, and arachidonic acid is thus produced, which is further converted into prostaglandin analogues via cyclooxygenase enzymes, which causes inflammation. Thus, kaempferol inhibits cyclooxygenase enzymes and prevents the inflammatory process. Kaempferol and cytokines have been incubated with isolated human hepatocytes and it was noted that COX2 and iNOS levels were decreased [6,45]. Lee and his coworkers found that kaempferol inhibits the expression of COX2 and lipo-oxygenase (LOX). It has been found that nitric oxide is involved in different complications such as inflammation. It has been reported that nitric oxide along with different metals deactivate catalase enzymes which in turn causes toxicity due to H_2_O_2_ accumulation. It has also been reported that proinflammatory cytokines like tumor necrosis factor alpha (TNF-α) are also produced in response to nitric oxide [46,47,48]. Different researchers have reported that kaempferol is responsible for the inhibition of nitric oxide synthesis which was induced by the administration of LPS to J77 cells and RAW264.7 cells and resulted in the reduction of inflammation. It was also demonstrated that the levels of TNF-α, nitric oxide and IL-1B were reduced by the treatment of kaempferol in diabetic neuropathy patients [49,50].

Reactive oxygen species (ROS) and reactive nitrogen species (RNS) are formed as a result of oxidative stress at the damaged cellular sites which causes damage to the cellular lipids, DNA and cellular proteins [51,52]. These reactive species which are produced as a result of the cellular damage and inflammation stimulate compensatory responses like the chemotaxis of neutrophils and macrophages to decrease the inflammation [53]. In vitro studies have shown that kaempferol is a good antioxidant agent. Different researchers have reported the free radical scavenging effect of kaempferol and also the inhibitory effect of kaempferol against lipid peroxidation [54,55]. Toxic chemicals inhibit or impair the function of nuclear factor (erythroid-derive 2)-like-2 (Nrf2) which further increases ROS that leads to oxidative damages. Saw and his coworkers reported that co-administration of kaempferol and H_2_O_2_ into HepG2-C8 cells resulted in the stimulation of Nrf2 and decreased ROS levels by the activation of heme oxygenase-1(HO-1) which prevented oxidative damage. It has also been reported that kaempferol is involved in the suppression of nitric oxide and iNOS levels which are the key mediators for oxidative stress leading to inflammation [36,56]. When studied in diabetic nephropathy, kaempferol produced marked suppression of inflammation via RhoA/Rho kinase regulation in NRK-52E (rat renal proximal tubular epithelial cells) and RPTEC (primary human renal proximal tubule epithelial cells) cells [57].

#### In Vivo Studies

Atherosclerosis is characterized by chronic arteriolar inflammation. The major risk factors responsible for atherosclerosis are increased oxidated LDL, an elevated cholesterol level and free radicals which may be due to diabetes, hypertension, infection or cigarette smoking. The inflammation associated with atherosclerosis is characterized by the long-term adhesion of monocytes to the arterial endothelium. The activation of endothelial monocytes stimulates the accumulation of leukocytes. The release of chemical mediators like growth factors and the proinflammatory cytokines like TNF-α and IL-1β stimulate the inflammatory process in atherosclerosis. These proinflammatory cytokines further initiate the release of adhesion molecules like E-selectin, P-selectin andICAM-1. It has been shown by earlier studies that E-selectin, P-selectin and ICAM-1 are upregulated in atherosclerotic patients. Nowadays, these adhesion molecules are considered as the clinical biomarkers for cardiovascular disease.

In vivo studies were conducted by Kong and his coworkers to explore the anti-inflammatory effect of kaempferol as an atherosclerosis remedy. They administered kaempferol for 10 weeks to rabbits. After the study, they found that the cholesterol level and arteriolar lesions of rabbits were markedly reduced. Kong and his coworkers found that the serum level of TNF-α, cytokines, leukocytes, IL-1β, Intracellular adhesion molecule-1 (ICAM-1) and E-selectin were significantly decreased after the treatment of kaempferol. Thus, they found kaempferol as a potential anti-atherogenic agent which prevents vascular inflammation [58]. Kim and his coworkers carried out a study on rats and demonstrated that kaempferol has a potential inhibitory effect on nuclear factor-kappa B activation which provides the cascade for the activation of RAGE and results in inflammation.

Free amino acids and carbohydrates interact to form advance glycation end-products (AGE). This process is stimulated by reactive oxygen and nitrogen species. AGE causes oxidative stress by damaging the cellular macromolecules. RAGE is the receptor for AGE. Upon stimulation of these receptors, it causes increased production of reactive oxygen species, reduction in antioxidant enzymes levels, activation of protein kinase C and synthesis of nitric oxide. It has also been reported that activation of RAGE results in oxidative stress via the stimulation of nuclear factor-kappa B (NF-ĸB) and redox-sensitive transcription factors. NF-ĸB is linked with oxidative stress which further activates COX-2, IL-6, TNF-α, IL-8 and iNOS which are responsible for inflammation. Thus, it was discovered by Kim and his coauthors that the anti-inflammatory effect of kaempferol was due to the inactivation of NF-ĸB via inhibition of NADPH oxidase [59]. Similarly, kaempferol showed significant effects of decreasing skin fibrosis and downstream oxidative biomarkers [60]. When tested in male rats with knee osteoarthritis (OA) in a model of ACLT-induced OA, in combination with apigenin, it strongly improved the OA [61].

### 7.2. Clinical Studies

Both the in vivo and in vitro studies have shown that kaempferol is a potential therapeutic agent for the management of inflammation. To demonstrate the clinical prospects of kaempferol, some nutraceutical clinical trials were performed on diseased subjects to test for the anti-inflammatory effects of kaempferol.

Hosseinpour and his co-researchers carried out a nutritional clinical trial in Type-2 diabetic patients testing for inflammatory biomarkers. They administered two groups of patients with a legume-free isoenergetic diet and non-soya legume-based beans diet (beans contain 14 mg/kg kaempferol as mentioned in Table 1. They found that the kaempferol-rich bean diet significantly reduced the inflammatory biomarkers like C-reactive protein (CRP), IL-6 and TNF-α [62].

Navarro and his coauthors also performed a nutritional-based clinical trial on healthy adults. Cruciferous vegetables (cauliflower, broccoli and radish) were investigated in a clinical trial to detect inflammatory biomarkers. Different cruciferous diets (cauliflower contains 270 mg/kg kaempferol, broccoli 30–72 mg/kg and radish 38 mg/kg kaempferol) were given to the individuals and it was observed after 14 days that IL-6 and IL-8 were significantly lowered [63,64]. Various case–control and cohort studies were performed in healthy individuals and diabetic patients to investigate the relationship between a kaempferol-rich diet (dried broccoli) and inflammation. In another clinical study, male smokers were evaluated following the consumption of the kaempferol-rich diet containing broccoli (72 mg/kg of kaempferol) for inflammatory biomarkers. It was found that the male smokers who received the broccoli diet for 10 days showed a reduction in TNF-α and IL-6 which are inflammatory biomarkers (Table 3) [64].

New innovative techniques use nanotechnology for coating the surface of certain chemical compounds to enhance their bioavailability and systemic absorption. Kaempferol can be encapsulated using a coating of nanoparticles of poly (lactic acid-co-glycolic acid) (PLGA) and polyethylene oxide-poly propylene oxide-polyethylene oxide (PEO-PPO-PEO). It was noted that alone kaempferol had poor bioavailability due to its low solubility and had a low anti-inflammatory effect as compared to the encapsulated kaempferol. However, further clinical trials should be conducted to evaluate kaempferol as a drug for the management of different medical problems [65].

## 8. Toxicity Profile of Kaempferol

The safety and toxicological profile of kaempferol was evaluated by different researchers and there was a conflict in their conclusions. Some researchers found kaempferol as antimutagenic while some found it to be genotoxic [66,67,68,69]. However, it has been shown that kaempferol has antioxidant effects while sometimes it acts as a pro-oxidant which performs a key role in the genotoxic effect [70]. Flavonoids reduce free radicals and form a phenoxyl radical by donating a hydrogen atom. When a second radical reacts with the phenoxyl radical, it acts as an antioxidant, while if the phenoxyl radical reacts with the oxygen species, then it acts as a pro-oxidant [71]. The pro-oxidant further reduces copper and iron ions which play a vital role in lipid peroxidation and hydroxyl radical formation. Certain studies have shown that the changes in antioxidant and pro-oxidant activities and levels of different enzymes were mediated by the pro-oxidant effect of kaempferol [72,73]. It has been reported by numerous researchers that the mutagenic effect of kaempferol is due to the conversion of kaempferol into genotoxic quercetin by the action of the CYP 1A1 enzyme [74,75].

In vitro studies have shown that kaempferol is carcinogenic and toxic, while these effects were not reported in in vivo screenings. A study was conducted by Takanashi and his coworkers, in which kaempferol was administered orally for 540 days and it was concluded that there was no elevation in the incidence of tumor. It was assumed that the low oral bioavailability of kaempferol prevented the genotoxic effect [76]. It has been reported that kaempferol is contraindicated with folic acid-deficient and iron-deficient patients as it reduces their cellular uptake and bioavailability. Kaempferol is also contraindicated in cancer patients who are on etoposide therapy, as it interferes with its bioavailability [77,78,79].

## 9. Conclusions

While reviewing the different remarkable attributes of kaempferol, it was revealed that many reports showed that kaempferol possesses anti-inflammatory activity. It has been shown to be a safe and efficacious natural dietary anti-inflammatory agent in both in vivo and in vitro settings. Kaempferol improved markers of inflammation.

Unresolved inflammation is currently ranked amongst the leading medical issues. Modern therapies are available to resolve this issue but most pose high health risks. Though kaempferol has improved several aspects of inflammation while posing minimal health risks, why it is still not available in clinical practices for the management of inflammation is a question needing an answer. The issue of the poor bioavailability of kaempferol has been resolved by nanotechnology. Kaempferol, as a natural compound, may elicit great variability in its therapeutic results. Numerous in vitro, in vivo and some clinical trials have been performed to investigate the anti-inflammatory potential of kaempferol. However, still further clinical trials must be carried out to confirm the worth of kaempferol in clinical settings for the management of inflammation.

## Figures and Tables

**Figure 1 molecules-25-04073-f001:**
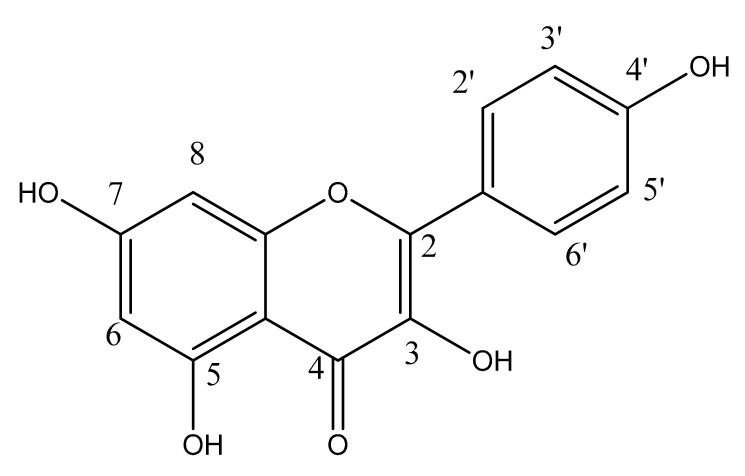
Chemical structure of kaempferol.

**Figure 2 molecules-25-04073-f002:**
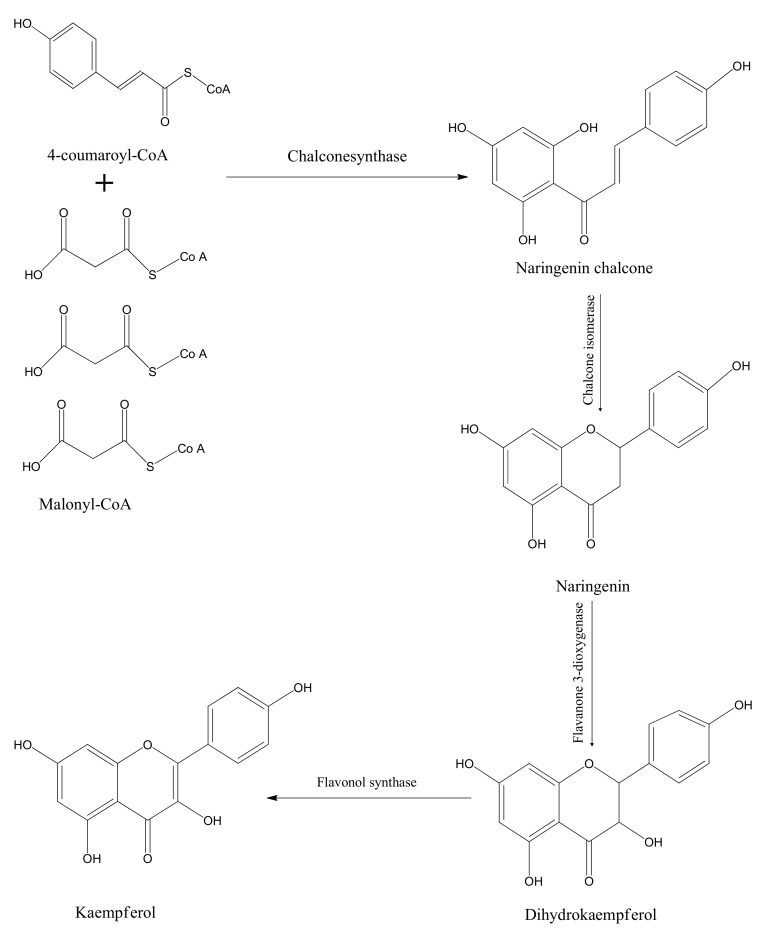
Biosynthesis of kaempferol.

**Table 1 molecules-25-04073-t001:** Edible and dietary sources of kaempferol.

	Food/Plant Beverages	Quantity	References
**Kaempferol**	Strawberry	5–8 mg/kg	[12]
	Gooseberry yellow	16 mg/kg	[12]
	Gooseberry red	19 mg/kg	[12]
	Onion leaves	832 mg/kg	[13]
	Black tea	118 mg/kg	[14]
	Green chili	39 mg/kg	[14]
	Papaya shoots	453 mg/kg	[14]
	Brinjal	80 mg/kg	[14]
	Pumpkin	371 mg/kg	[14]
	Carrot	140 mg/kg	[14]
	White radish	38 mg/kg	[14]
	Beans	14 mg/kg	[15]
	Broccoli	72 mg/kg	[15]
	Broccoli	30 mg/kg	[16]
	Cauliflower	270 mg/kg	[16]

**Table 2 molecules-25-04073-t002:** Cellular and molecular mechanisms of action of kaempferol as an anti-inflammatory agent.

	Mechanism of Action	References
**Anti-Inflammatory Effect**	Inhibits the NF-κB binding activity of DNA and myeloid differentiation factor 88	[38]
	Suppresses the release of IL-6, IL-1β, IL-18 and TNF-α.	[39]
	Increases mRNA and protein expression of Nrf2-regulated genes	[40]
	Inhibits the toll-like receptor 4 (TLR4)	[41]

**Table 3 molecules-25-04073-t003:** Clinical trials showing the anti-inflammatory effect of kaempferol.

Clinical Trials	Anti-Inflammatory Response	References
Type-2 diabetic patients with inflammation were treated with kaempferol-rich diet	Decreased the levels of inflammatory biomarkers (C-reactive protein (CRP), IL-6 and TNF-α)	[62]
Cruciferous diet (kaempferol-rich diet) was administered to patients	Recovered the inflammatory biomarkers like IL-6 and IL-8	[63]
Male smokers with inflammation were treated with kaempferol-rich diet (broccoli) for 10 days.	Reduced the TNF-α and IL-6 levels (inflammatory biomarkers)	[64]

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
