# Peer review of "Kaempferol as a Dietary Anti-Inflammatory Agent: Current Therapeutic Standing"

_molecules, 2020, doi:10.3390/molecules25184073_

Round 1

Reviewer 1 Report

The paper is improved but some questions still remain.

Are cruciferous vegetables like broccoli high in kaempferol? If so, why aren't they listed in Table 1?

It is still unclear what the kaempferol-rich diet referenced on line 209 consisted of.

The clinical trials section is still very unclear - e.g., how much kaempferol is in these diets? How do other components in these vegetables contribute to the positive effects.

Author Response

Dear Reviewer

I am highly thankful for keenly studying our article and positively suggesting some changes. I am sure the incorporations of these suggestions has greatly aided to the overall strength of our article.    

Reviewer 1.

The paper is improved but some questions still remain.

Are cruciferous vegetables like broccoli high in kaempferol? If so, why aren't they listed in Table 1?

Reply:  As per suggestions, the quantity of kaempferol in broccoli is added in table 1

It is still unclear what the kaempferol-rich diet referenced on line 209 consisted of.

Reply: The suggested changes have been made and kaempferol-rich diet is elaborated.

The clinical trials section is still very unclear - e.g., how much kaempferol is in these diets? How do other components in these vegetables contribute to the positive effects?.

Reply:  The exact quantities of kaempferol have been added in the respective diets as per suggestions. As the study was done with controlled feeding design and cruciferous diet was evaluated from tested cruciferous botanical sources additionally the diet selection was done on the basis of glutathione s-transferase genotype which minimized potential confounding due to other factors that may affect inflammatory status. However, it is a review article so we are bound to presented the results of research findings reported. As the effects of other components are not explored yet therefore, we can’t specifically mention other components.  

Reviewer 2 Report

The authors review the recent literature for the antioxidant and anti-inflammatory properties of the flavonoid kaempferol.  I would like to commend the authors in addressing almost all of the previous comments. The manuscript is now much improved, although a few grammatical issues were introduced as shown below.

Major comments:

Some of the paragraphs are too short (only 1, 2, or 3 sentences) and should be combined with an adjacent paragraph to improve the flow for the reader.

Minor comments: wording

Line 22: this review was -> this article is

Line 22: review of an updated literature review -> updated review of the literature

Line 52: includes -> include

Line 53: This review was aimed -> The aim of this review is

Line 67: Remove the words “as under”

Line 113: of was -> was

Line 117: microgram -> microgram per ml

Line 146: prevent -> prevents the

Line 147: cytokine have been incubated with the -> cytokines have been incubated with

Lines 152 and 181: pro inflammatory -> proinflammatory

Line 156: IL-B -> IL-1B

Line 168: decreased -> that decreased

Line 171: it -> kaempferol

Line 182: andICAM-1 -> and ICAM-1

Line 183: up regulated -> upregulated

Line 197: It has been reported that there are receptors (RAGE) -> RAGE is the receptor

Line 213: iso energetic -> isoenergetic

Line 225: are the -> are

Line 226: Nanotechnology is an innovative technique -> New innovative techniques use nanotechnology

Line 242: reduces the -> reduces

Line 254: interferes in -> interferes with

Line 256: remove the words “as an anti-inflammatory agent” [redundant with next line]

Line 256: it has been revealed -> it was revealed that many reports showed

Line 260: Inflammation -> Unresolved inflammation

Line 262: but why -> why

Line 263: questioned mark, needed to be answer -> question needing an answer

Author Response

Dear Editor/Reviewer

I am highly thankful for keenly studying our article and positively suggesting some changes. I am sure the incorporations of these suggestions has greatly aided to the overall strength of our article.    

Reviewer 2.

Major comments:

Some of the paragraphs are too short (only 1, 2, or 3 sentences) and should be combined with an adjacent paragraph to improve the flow for the reader.

Reply: As per suggestion, we have combined the said paragraphs and highlighted.  

Minor comments: wording

Line 22: this review was -> this article is

Reply: Needful correction is done and highlighted.

Line 22: review of an updated literature review -> updated review of the literature

Reply: Needful correction is done and highlighted.

Line 52: includes -> include

Reply: Needful correction is done and highlighted.

Line 53: This review was aimed -> The aim of this review is

Reply: Needful correction is done and highlighted.

Line 67: Remove the words “as under”

Reply: Needful correction is done and highlighted.

Line 113: of was -> was

Reply: Needful correction is done and highlighted.

Line 117: microgram -> microgram per ml

Reply: Needful correction is done and highlighted.

Line 146: prevent -> prevents the

Reply: Needful correction is done and highlighted.

Line 147: cytokine have been incubated with the -> cytokines have been incubated with

Reply: Needful correction is done and highlighted.

Lines 152 and 181: pro inflammatory -> proinflammatory

Reply: Needful correction is done and highlighted.

Line 156: IL-B -> IL-1B

Reply: Needful correction is done and highlighted.

Line 168: decreased -> that decreased

Reply: Needful correction is done and highlighted.

Line 171: it -> kaempferol

Reply: Needful correction is done and highlighted.

Line 182: andICAM-1 -> and ICAM-1

Reply: Needful correction is done and highlighted.

Line 183: up regulated -> upregulated

Reply: Needful correction is done and highlighted.

Line 197: It has been reported that there are receptors (RAGE) -> RAGE is the receptor

Reply: Needful correction is done and highlighted.

Line 213: iso energetic ->isoenergetic

Reply: Needful correction is done and highlighted.

Line 225: are the -> are

Reply: Needful correction is done and highlighted.

Line 226: Nanotechnology is an innovative technique -> New innovative techniques use nanotechnology

Reply: Needful correction is done and highlighted.

Line 242: reduces the -> reduces

Reply: Needful correction is done and highlighted.

Line 254: interferes in -> interferes with

Reply: Needful correction is done and highlighted.

Line 256: remove the words “as an anti-inflammatory agent” [redundant with next line]

Reply: Needful correction is done and highlighted.

Line 256: it has been revealed -> it was revealed that many reports showed

Reply: Needful correction is done and highlighted.

Line 260: Inflammation -> Unresolved inflammation

Reply: Needful correction is done and highlighted.

Line 262: but why -> why

Reply: Needful correction is done and highlighted.

Line 263: questioned mark, needed to be the answer -> question needing an answer

Reply: Needful correction is done and highlighted.

Round 2

Reviewer 1 Report

The authors have satisfied all my questions.

This manuscript is a resubmission of an earlier submission. The following is a list of the peer review reports and author responses from that submission.

Round 1

Reviewer 1 Report

As requested, I reviewed the manuscript (ID molecules-851119) "Kaempferol as a dietary anti-inflammatory agent: Clinical prospects", by W Alam, H Khan, A Shah, O Cauli and L Saso.

The manuscript deals with the ameliorative effects of natural drugs used in novel therapeutic strategies against inflammatory onsets and inflammatory diseases, with a direct focus on the flavonoid kaempferol. The main traits of kaempferol chemistry and biochemistry are described. Major attention is given to kaempferol anti-inflammatory actions, with a description of the related literature based on in vitro as well as in vivo research approaches and clinical studies.

The review manuscript is well conceived, with balance among the different sections.

Overall, the discussed issues contain all the elements for comprehension of the subject, for understanding the cited results and for finding hints to deepen information.

The reading sounds like a good and comprehensible continuum, and there’s no feeling of fragmented information. The manuscript is quite conceived for giving the necessary keywords to enter the investigated subject, and actually reaches this goal.

On this basis, the authors demonstrated scientific mastery of the research.

I only recommend a minor-but-fine revision of the English form throughout the entire document, for reaching a better final shape.

Reviewer 2 Report

Major comments: The review summarizes the anti-inflammatory actions and plant sources of the flavonoid kaempferol.  Most of this information provided can be found scattered throughout the literature in other review articles. The authors have assimilated this information from over 200 references and organized it in a logical manner. So, this review does provide some value to the literature. However, the English grammar and word usage are very poor and extensive improvements by a fluent English speaker/writer are needed. I have made extensive required changes as shown below.

Most clinical trials mentioned were with flavonoid-rich diets, not with pure kaempferol supplements, so the positive anti-inflammatory health effects cannot be ascribed directly to kaempferol. A table summarizing the clinical results would be helpful to clarify this point.

Minor comments:

English grammar and word usage

Line 3 (title): Clinical -> clinical

Line 14: Now a day’s -> Current

Line 14: well thought-out as the -> carefully chosen as a

Line 16: Phytochemicals -> phytochemicals

Line 16: In vivo and In Vitro -> in vivo and in vitro

Line 16: Among them -> Among them,

Line 18: which has the potency -> having the ability

Line 19: the fruits -> fruits

Line 19: Kaempferol is found in vegetables and food it is also found in the -> It is also found in

Line 20: Remove the words “and reported scientifically”

Line 21: Remove the words “as such”

Line 21: sources, -> sources and

Line 22: and specially focusing -> specifically focusing on

Line 23: paper was aimed to provide -> review was aimed at providing

Line 24, 26, 75, 77: Kaempferol -> kaempferol

Line 25: the available updated literature -> an updated literature review

Line 25: Natural -> natural

Line 32: the class -> a class

Line 33: kind -> kinds

Line 36: indispensable -> are indispensable

Line 39: Remove the word “systems”

Line 40: product -> products

Line 40: natural plants -> plants

Line 41: Remove the words “of plants”

Line 42: carbon ring -> the structure of their carbon rings

Line 43: which are -> and these divisions include

Line 43: majorly contributes -> are major contributors

Line 44: also helps in the -> helps

Line 45: effect -> effects

Line 45: proved -> shown

Line 47: oxides -> oxide

Line 49: tageretin, Luteolin -> luteolin [remove first instance of misspelled tangeretin]

Line 50: tangeritin -> tangeretin [keep second instance of misspelled tangeretin]

Line 50: sub group -> sub-group

Line 51: of flavonoids -> and of flavonoids

Line 51: ketone -> a ketone

Line 52: hydroxyl -> a hydroxyl

Line 52: ring -> central oxygen containing 6-member ring

Line 54: design an article which summarizes comprehensively -> comprehensively summarize

Line 55: Kaempferol majorly focusing -> kaempferol majorly focusing on

Line 56: effect -> effects

Line 59: the fruits -> fruits

Line 59: Kaempferol is found in vegetables and food it is also found in the -> It is also found in

Line 60: Remove the words “and reported scientifically”

Line 61: Remove the words “as such”

Line 62: sources, -> sources and

Line 62: and specially focusing -> specifically focusing on

Line 62: effect -> effects

Line 66: plant -> the plant

Line 67: are richly -> is richly

Line 76: coworker -> coworkers

Line 77: derivative -> derivatives

Line 78: Sephadex LH-20 scolumn -> a Sephadex LH-20 column

Line 78: co researchers -> co-researchers

Line 79: High speed counter current -> high-speed countercurrent

Line 80: leaves -> leaf

Line 80: two phase solvents -> a two-phase solvent

Line 82: MS, NMR -> MS,

Line 84: which are -> which is

Line 86: 4-coumaroyl-CoA -> 4-coumaroyl-CoA are used

Line 87: Chalcones -> Chalcone

Line 87: formation of -> formation of the

Line 88: enzyme Chalcone -> enzyme chalcone

Line 88: converts -> converts the

Line 88: flavonoid naringenin into flavanone naringenin -> naringenin chalcone into naringenin flavanone

Line 90: by the help of -> using the catalytic activity of the

Line 91: the help of flavonol synthase and kaempferol is formed -> flavonol synthase

Line 95: absorption -> Absorption

Line 96: new -> newly

Line 98: it is necessary for its therapeutic effects -> for its therapeutic effects it is necessary for it

Line 101: are lipophillic -> is lipophilic [misspelled]

Line 102: from -> from the

Line 104: active -> by active

Line 105: sulfate -> by sulfate

Line 106: enzyme -> enzymes

Line 106: colon -> the colon

Line 107: converted -> convert it

Line 108: systemic -> the systemic

Line 108: the distributed -> then distributed

Line 109: were -> was

Line 110: co researcher -> co-researchers

Line 111: human -> humans

Line 111: find out -> found

Line 111: of 57.86 nM was found -> was 57.86 nM

Line 112: about 15 ng/ml plasma concentration of kaempferol -> a plasma concentration of 15 ng/mL kaempferol

Line 114: different researchers -> several studies

Line 114: nano or micro gram -> the nano- or microgram range

Line 115: range -> effects

Line 116: This -> Thus,

Line 116: might me -> is

Line 117: the production of some active metabolites, -> some active metabolites produced

Line 124: immune -> the immune

Lines 125, 127, 129, 131: inflammation -> inflammatory

Line 126: Remove the word “system”

Line 126: Phosphodiesterase -> phosphodiesterase

Line 128: of protein kinase -> by protein kinases,

Line 130: via -> via the

Line 130: in -> in the

Line 132: effect -> effects [both instances (2 places) in this line]

Line 132 and 134: mechanism -> mechanisms

Line 133: is summarized as under -> are summarized below

Line 134: as -> as an

Line 135: In vitro studies -> In Vitro Studies

Line 136: Cyclo-oxygenase -> The cyclooxygenase

Line 136: the mediators -> mediators

Line 137: inflammation site -> site of inflammation

Line 139: causes -> cause

Line 141: cytokine has -> cytokines have

Line 141: Remove the words “change liver cell”

Line 142: level was found to be -> levels were

Line 142: coworker -> coworkers

Line 143: inhibit -> inhibits

Line 143: evaluated -> found

Line 144: like inflammation process -> such as inflammation

Line 144: is -> has been

Line 145: accelerate different -> deactivate

Line 146: is also -> has also been

Line 146: Tumor -> tumor

Line 146: a pro inflammatory cytokine -> proinflammatory cytokines

Line 147: is also produce by -> are also produced in response to

Line 148: LPS in -> the administration of LPS to

Line 149: results -> resulted

Line 149: It is -> It was

Line 150: is -> were

Line 150: the diabetic neuropathic -> diabetic neuropathy

Line 155: the mechanistic pathway like -> compensatory responses like the

Line 155: subside -> decrease

Line 157: has been -> is

Line 158: inhibiting -> inhibitory

Line 159: impaired -> impair

Line 160: causes the over expression of -> increases

Line 160 and 162: Nrf-2 -> Nrf2

Line 161: co administration -> co-administration

Line 162: results -> resulted

Line 162: which further reduces the synthesis of ROS -> decreased ROS levels

Line 163: Haemooxygenase-1 -> heme oxygenase-1

Line 163: prevents -> prevented

Line 164: are involved -> is involved

Line 164: level -> levels

Line 167: epithelial cell -> epithelial cells

Line 168: In Vivo Studies -> In Vivo Studies [Italicize]

Line 169: Currently atherosclerosis is well accepted and -> Atherosclerosis is

Line 172: adhesion -> the long-term adhesion

Line 173: with arterial endothelium for long term -> to the arterial endothelium

Line 173: endothelium -> endothelial

Line 173: stimulates -> stimulates accumulation of leukocytes

Line 174: cytokines, growth factors, -> growth factors and

Line 174: Remove the words “accumulation of leukocytes and”

Line 175: progress -> stimulate

Line 177: (ICAM-1) -> ICAM-1

Line 177: proved -> shown

Line 177: (ICAM-1) -> and ICAM-1

Line 180: co researchers -> coworkers

Line 181: as -> as an

Line 183: Remove the words “Progression of”

Line 186: Kaempferol as the potential -> kaempferol as a potential

Line 187: coworker -> coworkers

Line 188: potential -> a potential

Line 188: the nuclear -> nuclear

Line 191: the Reactive species -> reactive oxygen and nitrogen species

Line 191: the cellular physiology -> cellular macromolecules

Line 192: AGE upon -> AGE. Upon [Break into 2 sentences]

Line 193: caused the activation of Reactive Oxygen Species -> causes increased production of reactive oxygen species

Line 193: enzymes level -> enzyme levels

Line 194: production -> activation

Line 194: Remove the word “synthase”

Line 195: the oxidative -> oxidative

Line 196 and 197: Remove the words “the chemical mediators like”

Line 198: figure out -> discovered

Line 199: Oxidase -> oxidase

Line 199: it -> kaempferol

Line 200: skin -> decreasing skin

Line 200: via downstream regulation oxidative biomarker -> and downstream oxidative biomarkers

Line 200: rat with knee osteoarthritic in -> rats with knee osteoarthritis (OA) in a

Line 202: Remove the words “cell therapy”

Line 206: done on disease models for kaempferol anti-inflammatory -> performed on diseased subjects to test for anti-inflammatory effects of kaempferol

Line 207: co researchers -> co-researchers

Line 207: patient -> patients testing

Line 208: iso energetic -> isoenergetic

Line 209: Kaempferol rich -> kaempferol-rich [both instances]

Line 209: diet has -> diet

Line 212: clinical -> a clinical

Line 212: the inflammatory -> inflammatory

Line 214: Remove the word “down”

Line 216: kaempferol rich -> a kaempferol-rich

Line 217: against -> following the consumption of

Line 217: kaempferol rich -> kaempferol-rich

Line 217: like -> containing

Line 218: received -> received the

Line 221: Remove the words “confirm its”

Line 221: is shielded by capsule -> can be encapsulated using a

Line 222: ploy ethylene -> polyethylene

Line 224: low -> its low

Line 224: has poor-> had poor

Line 224: has low potential-> had a low

Line 226: Remove the word “still”

Line 226: evaluated the kaempferol as clinical -> evaluate kaempferol as a

Line 230: found a conflict in their conclusion -> a conflict in their conclusions

Line 231: thought it as -> found it to be

Line 231: was proved -> has been shown

Line 231: effect -> effects

Line 232: as -> as a

Line 232: perform key role in -> performs a key role in the

Line 232: Flavonoid reduces a free radical -> Flavonoids reduce free radicals

Line 233: hydrogen -> a hydrogen

Line 233: react with -> reacts with the

Line 234: antioxidant -> an antioxidant

Line 234: phenoxyl radical react with oxygen specie then it acts as -> the phenoxyl radical reacts with the oxygen species then it acts as a

Line 235: reduce -> reduces

Line 235: vital -> a vital

Line 236: radicals -> radical

Line 236: antioxidant -> changes in antioxidant

Line 238: researcher -> researchers

Line 240: are -> is

Line 241: screening -> screenings

Line 241: coworker -> coworkers

Line 242: were -> was

Line 243: This -> It

Line 243: kaempferol low oral bioavailability prevents -> low oral bioavailability of kaempferol prevented

Line 246: in -> with

Line 248: as -> as an

Line 248: agents -> agent

Line 249: potential anti-inflammatory effect -> anti-inflammatory activity

Line 249: proven a safest -> shown to be a safe

Line 250: In vivo and in vitro -> in vivo and in vitro [italicize]

Line 251: resolved all the aspects responsible for -> improved markers of

Line 252: issue -> issues

Line 253: the issue but they -> this issue but most

Line 253: Kaempferol -> kaempferol

Line 253: resolved all -> improved

Line 254: with less -> while posing minimal

Line 254: still -> why it is still

Line 255: questionable? -> questioned.

Line 257: done -> performed

Line 258: execute -> confirm

Line 260: supported of -> support from

Line 261: review -> the review

Line 263: Remove the words “Please add:”

Reviewer 3 Report

General Comments 

This paper investigates the evidence for the beneficial effects of kaempferol. Suggestions for improvement are detailed below.

1) The grammar and English in the manuscript is poor, and it makes most of the paper unreadable. The paper should be edited by a native English speaker.

2) The title does not really capture what the review is about – the manuscript does not focus on clinical prospects but rather the evidence for kaempferol’s anti-inflammatory properties.

3) Table 1 is not useful for the overall review. If kept, it should be moved to the supplement. More useful would be a table with some foods or beverages that contain kaempferol and in what quantities. After all, the title says “dietary”, and after reading the paper I still have no clue how to obtain kaempferol from the diet.

4) Is kaempferol a flavonoid? If so, why are flavanols discussed on lines 51-52?

5) On line 116, change “me” to “be”.

6) The in vitro section of the manuscript (7.1) reads like a long list of descriptions with no discussion on what any of it means. As an example, what does it mean to inhibit COX2?

7) It is unclear what the kaempferol-rich diet referenced on line 209 consisted of.

8) When discussing the clinical trials on lines 211-219, it is unclear how much kaempferol is in these diets, or how other components in these vegetables might be contributing to the positive effects.

9) The statement on line 251 that “Kaempferol resolved ALL the aspects responsible for inflammation” cannot even begin to be true, as all aspects cannot be measured. This should be rephrased.